# Development and Evaluation of a Virtual Model for Fetal Alcohol Spectrum Disorder (FASD) Assessment and Diagnosis in Children: A Pilot Study

**DOI:** 10.3390/children10020196

**Published:** 2023-01-20

**Authors:** Seema King, Colleen Burns, Brent Symes, ShawnaLee Jessiman, Amber Bell, Hasu Rajani

**Affiliations:** 1Department of Community Health Sciences, Cumming School of Medicine, University of Calgary, Calgary, AB T2N 1N4, Canada; 2Lakeland Centre for Fetal Alcohol Spectrum Disorder, Cold Lake, AB T9M 1Y6, Canada; 3Communicating Together Inc., High Level, AB T0H 1Z0, Canada; 4Department of Pediatrics, Faculty of Medicine & Dentistry, University of Alberta, Edmonton, AB T6G 2R3, Canada

**Keywords:** fetal alcohol spectrum disorder, fetal alcohol syndrome, prenatal alcohol exposure, neurodevelopment, virtual care, pediatrics

## Abstract

The diagnostic process for fetal alcohol spectrum disorder (FASD) involves a multi-disciplinary team and includes neurodevelopmental, physical, and facial assessments and evidence of prenatal alcohol exposure during the index pregnancy. With the increased use of virtual care in health care due to the pandemic, and desire of clinics to be more efficient when providing timely services, there was a need to develop a virtual diagnostic model for FASD. This study develops a virtual model for the entire FASD assessment and diagnostic process, including individual neurodevelopmental assessments. It proposes a virtual model for assessment and diagnosis of FASD in children and evaluates the functionality of this model with other national and international FASD diagnostic teams and caregivers of children being assessed for FASD.

## 1. Introduction

Over the last decade, there has been a dramatic shift towards physicians providing care virtually, further increased by the COVID-19 pandemic. Virtual care is defined as “any interaction between patients and/or members of their circle of care, occurring remotely, using any forms of communication or information technologies with the aim of facilitating or maximizing the quality and effectiveness of patient care” [1]. Prior to the pandemic, the role of virtual care, mostly by telehealth, was accepted as a beneficial and effective method for assessing, diagnosing, and communicating with patients, with the aim of facilitating or maximizing the quality and effectiveness of their care [2].

Virtual care can reduce wait times for those living in rural and remote regions [3,4,5]. Badawy and Radovic [6] highlight the digital options for these communities as ways to “increase access to effective, accessible, and consumer-friendly care for more patients and families.” In Alberta, Canada, over 16% of the population is below 18 years of age and over half a million inhabitants live in rural and remote areas [7]. Patients living in these communities face challenges, not only in access to general medical care, but also to specialized health care and specialists including those in pediatric care. These challenges result in missed appointments, and care being accessed through emergency departments when health conditions become more acute [8]. Lack of access to primary and specialty care contributes to increase in disparities in health outcomes [8].

Fetal alcohol spectrum disorder (FASD), resulting from prenatal alcohol exposure (PAE), is a condition resulting in variable expression of physical and lifelong neurodevelopmental changes and outcomes for the individual [9]. Currently, in Alberta, there are over 20 diagnostic teams in a variety of settings including urban, rural, remote, and hospital-based clinics [10]. Several clinics have team members who travel to a central site to complete assessments. The caregiver and child also travel from their home community to the site where assessment and diagnosis is scheduled and completed. Barriers to attendance include weather conditions, expenses, clinicians’ schedules, and long travel distances to sites.

The diagnostic process for FASD is complex and the Canadian Guideline for FASD Diagnosis recommends a multidisciplinary team for assessment [11]. The core diagnostic team usually consists of a physician or nurse practitioner, a speech language pathologist (SLP), a psychologist, and an occupational therapist (OT). Team members from other disciplines such as education, justice, social services, mental health, and cultural liaison workers complement the full multidisciplinary team.

The aim of this pilot study was to improve access to pediatric FASD diagnosis and assessment through the development of a virtual model. Secondary aims included exploration of use of the model with national and international clinicians, specializing in FASD assessment and diagnosis, and survey evaluation of caregiver experiences using the model.

## 2. Methods

A virtual model of FASD assessment for children based on validated assessment tools was developed building upon a previous literature review of FASD assessments [12]. After development, this model was presented in webinars to other FASD assessment and diagnostic teams provincially in May 2020 and internationally in June 2020. Participants were surveyed post-webinar about their current use of virtual methods in FASD assessment and diagnosis as well as assessments they would consider completing virtually after attending the webinar about the virtual model.

The virtual model was also piloted in practice in four pediatric FASD assessment clinics in Alberta, Canada from June 2020 to June 2021. Clinic coordinators in each clinic consented caregivers to fill out a survey evaluation after a virtual FASD assessment for their child. In this study, caregivers were defined as the individual who assumes the role of providing day-to-day care for a child, including parent(s), foster parent(s), kinship care providers, or other authorized individual(s). The pilot focused on using the virtual model with children aged 7–16 years of age with no severe behavioral issues, and the caregivers’ ability to complete questionnaires and provide consent. School-aged children were chosen because they are often the ones identified as requiring cognitive testing, are readily adaptable to learning and using technology, and are less likely to have face-to-face behavior management issues.

The evaluation focused on the virtual aspect of the assessment process and included questions assessing the caregiver’s understanding of the process, the quality of the method, the caregiver and child’s comfort level, their satisfaction, and the acceptability of using a virtual model. The survey was conducted in two rounds, one with a small group of caregivers from June 2020 in one clinic expanding to a larger evaluation in three clinics from April to July 2021. Changes were made to the survey between iterations, but questions that remained the same were grouped together in analysis. Respondents were asked to rate their agreement to statements using a 5-point Likert scale and were provided open text comments to expand on answers that were unsure or on the lower end of the scale.

Ethics approval (25 June 2020) was obtained from the University of Alberta Research Ethics Board (Pro00101938 and Pro00101860).

## 3. Results

### 3.1. Development of a Virtual Model

#### 3.1.1. In-Person FASD Diagnosis and Assessment Model

Currently, most of the process for diagnosis and assessment of FASD is completed in person and follows the steps outlined in Figure 1.

The first step is *referral and intake,* where a referral, usually originating from frontline workers (such as social workers, teachers, etc.) or family is accepted, as per each clinic’s screening criteria (i.e., age of referral; pre-screening or pre-assessment required). A clinic coordinator is responsible for the referral, intake, information collection, PAE confirmation, and medical summary report compilation and distribution. Consent forms for the access and release of information are signed by the legal guardians, allowing clinics to obtain health records (including birth records), education reports, child and family services history and information, justice and other relevant reports and documentation, including results of any previous assessments. During this time, reliable and accurate details of PAE are collected in a respectful and non-judgmental manner, in accordance with the Canadian diagnostic guideline recommendations [11]. Assistance is provided, as needed, to legal guardians, caregivers, and referral sources to complete the intake forms. Caregivers and teachers are requested to complete questionnaires (i.e., adaptive behavior and related skills) prior to the assessments by the clinicians.

Once the referral process is complete, *multidisciplinary assessments* are scheduled by the clinic coordinator to be completed by core team clinicians, including a physician, registered psychologist, speech language pathologist and occupational therapist. Parent and teacher questionnaires and rating scales are completed, with the clinic coordinator assisting with this process. Any previous speech–language, occupational therapy, educational, psychiatric, neurodevelopmental, and other pertinent assessments are obtained and reviewed with respective clinicians to ensure history and validity of previous testing protocols (i.e., appropriate time frame between testing).

The physician (or nurse practitioner) conducts an *interview with the caregiver* to support and augment information and any history obtained. This is a standard practice in FASD clinics and informs the physician of the caregiver’s concerns, understanding, and expectations of the assessment process. The physician explores the presence and frequency of PAE associated behaviors and clarifies any medical information previously collected. The child is also interviewed, and when appropriate, an adolescent HEADSS history is completed, ensuring confidentiality is maintained [13]. An older child is asked regarding their understanding of the purpose of the visit. The interview with the child allows for observation of any significant dysmorphic features and listen for speech or language issues. A partial neurological exam including deep tendon reflexes, tone, Romberg test, tandem gait and dysdiadochokinesia is performed. The child is observed for their ability to jump or hop on alternate feet. Observations of quality of eye contact, patient’s affect, activity level, and response to instructions and commands provide additional information. The physician completes a physical examination to assess for physical anomalies, and growth measurements, including height, weight, and head circumference. The three sentinel facial features are assessed by measuring the palpebral fissures, on each side, using a ruler, while the University of Washington Lip-Philtrum Guide is used to evaluate the upper lip and philtrum development [14].

Once assessments are complete, a *multidisciplinary case conference* for diagnostic formulation, and recommendations is scheduled. At this time, the team members review the case history, and other information collected through interviews, reports, and assessments. Each clinician reports their findings on test measures performed. The members discuss and deliberate on the significance of the test results, the degree of impairment of the different domains of brain function and determine whether the criteria for FASD diagnosis and other diagnoses are met. Based on findings, a set of recommendations that will guide post-clinic interventions and services are developed to ensure support for the child and the caregiver.

The final step of this process is the *caregiver/patient debrief*. The physician (or nurse practitioner) and other core clinicians communicate the diagnosis to caregivers. The patient may be involved in the debrief process where appropriate and after discussion with the caregiver. A strength-based approach is followed when reviewing assessment results and recommendations with caregivers and the child (dependent on the child’s age and direction of legal guardians/caregivers). On the clinic day, a brief medical report including the diagnosis and recommendations is prepared and given to the family after the debrief, with a final detailed comprehensive report provided a short time later.

#### 3.1.2. Virtual Assessments

There are few studies that focus on virtual models to complete the full battery of neurodevelopmental and speech–language assessments; caregiver and patient interviews; multidisciplinary diagnostic formulation; and caregiver and patient debriefing sessions specific to FASD assessment and diagnosis [12]. Pradhan and Six-Workman discussed a telehealth model for integrated mental health services in a rural school-based health clinic for the vulnerable youth, and described barriers, challenges, and efficient use of resources [15]. Three past studies explored “the viability, effectiveness, and experiences of telehealth programs specific to FASD” and evaluated the FASD Telehealth program within two rural and remote Manitoba communities [16,17,18]. Challenges around technology, such as picture clarity and consistent, reliable connectivity between the patient and medical team were identified, as well as the need of further research to understand the perspectives of individuals and families who accessed the services through telehealth. None of these studies completed all aspects of the FASD assessment and diagnostic processes using a virtual model but other studies have used virtual methods reliably for specific assessments.

Speech–language assessment: A comprehensive speech–language assessment is important in the FASD diagnostic process. Sutherland and Trembath used telehealth to assess 23 school-aged children with known or suspected language impairments [19]. The results showed good agreement on all test measures, and feedback from caregivers and the youth was positive and supportive. In the proposed virtual model, the SLP uses a core battery of assessment tools consistent with those outlined in the Canadian Guideline for FASD diagnosis. These are the same tools that were used during face-to-face assessments (Appendix A). The SLP assessment contributes to the evaluation of three brain domains: Language, Social Communication and Adaptive, as well as Executive Function. The virtual platform is designed to minimize the training required for a support worker, who is in the room with the child being assessed, to monitor testing behavior, help with technology, and keep the participant focused and on task as requested by clinician. A final assessment of the results of the tests completed by the SLP is made using evidence-based practice. In Alberta, as directed by the Alberta College of Speech-Language Pathologists and Audiologists, the current test battery’s standardized test scores cannot be reported as virtual testing is an adaptation of the testing presentation. The clinician’s experience, scientific data based on comparable tele-practice versus face-to-face results, and information from the patient’s school and families are used in determining the speech and language diagnosis [19,20,21].

Motor skills assessment: An FASD assessment includes an assessment of motor skills by an occupational therapist as children with FASD and PAE have been shown to have deficits in motor skills [22]. Although telehealth is used in occupational therapy practice, studies have not supported a reliable virtual assessment of motor skills [23]. Hen-Herbst, Jirikowic compared children with PAE and Developmental Coordination Disorder, noting only 1 of 21 children showed impairment in motor skills below the 5%, whereas 13 showed mild and 7 showed moderate difficulties on the Movement ABC (MABC-2) instrument [24,25]. In a retrospective study of chart reviews and testing performance on motor skills comparing assessment tools MABC-2 and BOT-2SF and the BeeryVMI-6 and BeeryMC, Johnston and Branton noted that the MABC-2 was most sensitive and found impairment in 30% of children with FASD [26]. However, this has not been replicated in other studies. Lucas and Doney noted that 2 of 21 (9.8%) children with an FASD diagnosis showed motor skills impairment below the third percentile [22]. Due to reports in studies of infrequent findings of significant impairment (below 3%) in motor skills with in-person testing, and that virtual assessment in this area has not been validated, a virtual assessment of this domain was not included in the virtual model for assessment of FASD. However, we recommend an in-person OT consultation post-clinic to address any concerns regarding motor skills, sensory-processing issues, and ADHD-related behaviors.

Neurodevelopmental assessments: Neurodevelopmental assessments require objective assessments of the various domains of brain function in FASD diagnosis [11]. Temple, Drummond compared the results of in person vs. virtual assessments using the WASI and Beery–Buktenica Development Test of Visual Motor Integration, concluding that the assessments were comparable [27]. Hodge, Sutherland compared in-person vs. virtual modes of assessment of intellectual functioning using the WISC-V in 33 children with a specific learning disorder [28]. The study results showed a high correlation for each of the subdomains and the full-scale IQ. Munro Cullum, Hynan studied 202 adults presented with a variety of neurodevelopmental tests and revealed strong correlation between results of in-person and virtual testing [29]. Limitations included using a short battery of tests and assessment measures that are appropriate for evaluation of presence of dementia. In this proposed FASD model, the psychologist completes and contributes to the evaluation of seven brain domains via a computer, with help from a support worker. The direct psychometric tools used for the evaluations included (Appendix B) are dependent on the age of the patient. Testing materials/protocols are compiled by the psychologist and placed into sealed envelopes and distributed directly to the clinical coordinators. Support workers are asked to keep the envelopes sealed until the beginning of each assessment. Once the evaluation has concluded, test materials/protocols are placed back into the envelopes and sealed for return to the psychologist.

Physical measures and neurological assessments: FASD assessment requires the completion of a physical exam for sentinel facial features, measurement of the patient’s head circumference, and a neurological examination by a trained individual. A piece of evidence-based software, for measurements of the sentinel facial features, is used to complete this by a team member with appropriate training. The University of Washington Fetal Alcohol Syndrome Diagnostic & Prevention Network (FAS DPN) FAS Facial Photographic Analysis Software is used to measure the magnitude of expression of the three key diagnostic facial features of FAS (short palpebral fissure lengths, smooth philtrum, and thin upper lip) [13]. The software was developed by Susan Astley Hemingway, PhD in 2003 (updated in 2012 and 2016) for use by health care and research professionals. It has been used to accurately measure the full continuum of expression of the FAS facial phenotype in thousands of individuals, from birth to adults, and has been evaluated in the FASD screening and diagnostic programs. In the proposed model, pilot site clinic coordinators and support persons were trained to take accurate photos and conduct analysis of photos using the software. This training included two interactive live demonstration and practice sessions, as well as the comparison of manual and software generated measurements. Proficiency using the software was gained through additional practice sessions using volunteer participants.

To assess for the presence of FAS facial phenotypes, the coordinator arranges for in-person photographs of the child to be taken either before or after the child’s virtual assessment. Three photographs of each child (frontal view, ¾ view, and lateral view) are taken and analyzed using the University of Washington facial software. The software generates a detailed report, scoring outcomes of the facial measurements using the 4-Digit Diagnostic Code. In consultation with the multidisciplinary team physician, these outcomes are reviewed to determine the presence (or absence) of the three FAS facial phenotypes assessed. The same team member is trained to carry out the measurements for height, weight, and head circumference and extrapolate the percentiles.

Abnormal neurological signs contribute to informing the domain of motor skills, which is assessed in major part by the occupational therapist. A reliable physical neurological exam cannot be conducted virtually; however, gross impairment in balance and some easily performed cerebellar tests can be completed virtually. Fitzpatrick, Latimer, in noting a prevalence rate of microcephaly that was nine times higher in children with FASD than amongst those with no PAE, did not find any difference in neurological signs between the two groups [30].

#### 3.1.3. Other Considerations of Using the Virtual Model

*Support worker:* A support worker is required when using a virtual model. This support worker can be from a local community agency or school, or be staff member of the FASD clinic who has received appropriate training in the use of virtual platforms and the necessary equipment and technology. The individual is tasked with setting up the necessary equipment and maintaining test/instrument confidentiality. Prior to the assessment, the psychologist and/or SLP and support worker initiate the virtual meeting to test the equipment before the patient is introduced. During the process, the support worker is responsible for monitoring the patient and ensuring images presented on the computer screen are clearly visible. At various points throughout the assessment, the psychologist may request the support worker to provide and collect documents/worksheets to and from the patient. With experience, the support worker may also provide comments on their observations of the patient during testing. This would provide additional useful information to the clinician conducting the test.

*Ethics:* Virtual care must conform to the rules and regulations of the governing and professional regulatory bodies. It is essential that the caregiver and patient, where appropriate, provide informed consent to the virtual process of assessment. The diagnostic team should be aware of the security capabilities of the virtual platforms to ensure patient confidentiality. The physical environment should enable assessments in a comfortable, quiet, private, and safe location allowing for confidential data gathering, interviewing, and testing.

*Equipment:* Equipment and connectivity should allow for a clear, continuous, and uninterrupted audio-visual connection. Ancillary equipment such as additional cameras with appropriate resolution should be used to ensure clear observation of the patient and testing materials. Training of the support worker for use of facial photographic software, taking appropriate facial photographs, measurement of the sentinel facial features, and growth parameters and participating in caregiver interviews, or supporting clinicians to present test materials to the patient, should be completed and practiced for proficiency. These members may also help the patient navigate equipment use and report issues of connectivity.

#### 3.1.4. Moving from an In-Person to Virtual Diagnostic Process

In addition to assessments, as described previously, there are several other components to the multidisciplinary assessment of children with FASD (Figure 1). These include gathering relevant information before the clinic, scheduling assessments to be completed by the clinicians, completing physical measurements, scheduling a team meeting for the discussion of results and the formulation of a diagnosis and recommendations, and a final meeting to discuss the results of the assessment with the caregiver (and patient).

The “referral and intake” process including records/information gathering by the clinic coordinator continues to be conducted electronically (i.e., by phone or other means of connecting with the referral source). When deciding on a virtual assessment, the criterion for accepting a pediatric referral for virtual care is discussed and consensus reached with all clinicians completing the testing. Children with severe behavioral issues or a preschool referral may not be appropriate for virtual assessment. Reports are obtained electronically to obtain accurate PAE documentation, and interviews with birth parents or other reliable sources are conducted by telephone or using a virtual platform. Similar to in-person interviews when collecting this sensitive information, the interviewer should be skilled and trained in motivational interviewing and have a good understanding of trauma-informed practice, especially when speaking with birth families. Virtual assistance is provided to caregivers completing the intake forms, as well as providing a clinic contact for any follow up or questions. Consent forms to access or disclose personal and health information are adapted appropriately when using electronic communication.

In the virtual model, the clinic coordinator fully explains the intent and process of the telehealth project, and ensures the legal guardian is well-informed prior to signing consent forms. It is important that the legal guardians understand the process and any risks associated with virtual connectivity, while understanding the security, privacy, and confidentiality measures that are in place.

The multidisciplinary case conference, diagnostic formulation, and recommendations are scheduled for the clinic team as soon as possible after the assessments are completed. Logistics for members to connect are organized and shared, such as any passwords required, and finding an appropriate meeting space in the community if multiple members are using the same location. The caregiver–physician interview may take place prior to the team meeting, or this can occur on the same day as the case conference before the team meets. Virtual connections enable community team members to attend the multidisciplinary meeting, allowing individuals to join without taking time from work or having to travel. Representation from education, child and family services, intervention service providers, and other team members attend virtually and contribute to the case review, diagnostic formulation, and recommendations.

The clinician(s) debriefing with the caregivers and patient (dependent on the age of the child) is conducted using a virtual mode that is pre-arranged and fully explained to the caregivers. Security and confidentiality are essential, and the caregivers are supported throughout to ensure this meeting occurs in a safe and private space. The debrief is often conducted on the same day as the case conference and diagnostic formulation day or can be scheduled for a date soon after. A short version of the medical summary report, including FASD and other diagnosis, along with recommendations, can be sent electronically through a secure method compliant with standards of practice within each discipline. The full medical summary report should be shared with legal guardians at a date soon after the clinic day. A member of the diagnostic team is responsible for following up with caregivers and ensuring access to support recommendations listed in the final medical summary report.

The FASD evaluation process, whether in-person or virtual, may need to occur over several appointments, depending on the comfort of the caregiver and/or patient. Comprehensive assessments can take 45–60 min each, which can be difficult for some children to sit through. Breaks, monitoring signs of disinterest, and spacing of assessments may be required. Not all steps for the virtual model (Figure 1) need to be completed virtually but can be completed by a combination of in-person, over the phone, or virtually depending on feasibility and patient or caregiver preference.

### 3.2. Evaluation of the Model

#### FASD Assessment and Diagnostic Teams

Webinars were conducted provincially in May 2020 with 24 attendees and in June 2020, with over 200 attendees. Of those that identified their professional role (*n* = 71), 13% were physicians, 41% were registered psychologist/psychometrists, 9.9% were speech language pathologists, 28% were clinic or network coordinators, and 9.9% marked other. Prior to the webinar, 40% (*n* = 82) were members of clinics that were already using virtual or telehealth platforms for parts of FASD assessment and diagnosis and 57% (*n* = 81) were considering use of virtual options. After the webinar, 77% (*n* = 71) were likely or very likely to consider the use of virtual options for at least one part of the FASD diagnosis and assessment process. Pre-webinar, many clinics of attendees had implemented virtual methods for conducting diagnostic formulation and recommendations by the clinic team, as well as debriefs with the caregiver, while post-webinar, most parts of the diagnostic process were being considered virtually by attendees (Figure 2).

Participants were also asked after the seminar if they would consider completing neurodevelopmental assessments virtually. Although some clinics were already using virtual methods for some neurodevelopmental assessments, many attendees considered using virtual methods for other neurodevelopmental assessments after the webinar (Figure 3).

### 3.3. Caregiver Experiences

The first round of surveys conducted in June 2020 had 6 respondents, and the second round from April–July 2021 had 27 respondents complete the full survey. One survey in the second round was excluded due to being less than 50% complete.

Parts of the assessment completed virtually varied for respondents (*n* = 27) with some having more than one part of the process completed virtually. The majority of caregivers (93%) received the diagnostic formulation and recommendations by the clinic team and the debrief of diagnosis with the caregiver virtually, while 63% had the interviews with the physician completed virtually. A total of 67% of caregivers had the neurobehavioral assessment completed virtually, and 19% had a virtual speech–language assessment. Prior to the appointment, 100% of respondents thought the process and details of connecting and using virtual technology were made clear to them, and 96% of them felt that they had adequate opportunity to ask questions about their appointments. The majority of caregivers strongly agreed or agreed favorably to aspects of using the virtual method including maintaining privacy, comfort with use, and quality of visual and audio (Figure 4).

Overall, 91% of caregivers (*n* = 33) were very satisfied/satisfied with the quality of services provided with virtual technology and 9% were not sure. The three respondents that were unsure commented that they were satisfied with the services but would have preferred it to be in-person if possible. A total of 77% of caregivers (*n* = 26) strongly agreed/agreed that virtual methods were an acceptable way to receive an assessment and case conference, while 15% were unsure. The two respondents that disagreed preferred an in-person assessment. A total of 82% of caregivers (*n* = 33) would have a virtual appointment again, with those who were unsure or declined to answer commenting that they would only do so if face-to-face was not an option.

The survey also asked how the virtual option benefited the caregiver. A total of 63% of caregivers (*n* = 27) selected that it saved travel time, 55.6% costs savings, 33.3% no missed work time, 33.3% ease of childcare arrangements, and 33.3% increased comfort connecting from home or another setting in their community. When asked if the quality of virtual services is comparable to the quality of in-person or face-to-face services: 52% strongly agreed or agreed, 18.5% were neutral, and 29.6% disagreed or strongly disagreed. Those that disagreed or strongly agreed noted that personal connection would have been more beneficial.

## 4. Discussion

The proposed virtual model of FASD diagnosis and assessment was developed and composed of assessments that are validated for virtual use. In addition, the use of this model was surveyed by national and international FASD assessment and diagnostic teams and piloted in practice. This model provides an evidence-based alternative to in-person assessments and provides guidance to pediatric FASD assessment and diagnostic teams that may need to rely on virtual methods to assess children, especially those located in remote and rural areas.

A virtual model for FASD assessment and diagnosis is a reliable model of assessment that has several advantages. For caregivers in our study, it saved travel time and costs, and decreased work time missed. It allows for the delivery of care in the patient’s community or closer to home, resulting in time and monetary savings for the patient and care providers, decreasing the number of missed appointments, and increasing the ability to engage community partners (e.g., schools). Specific to FASD, assessments for neurodevelopmental and language are reliable, and the validity of these tests is equivalent to those conducted face-to-face.

Previous studies have shown the benefits to virtual care to include timely access to care, reduced travel time and wait times, and improved convenience for patients [31]. These efficiencies were also shown to result in cost savings by reducing personal travel expenses and travel subsidies, and decreasing hospital service utilizations and transfer to tertiary care centers [31]. Virtual care can be easy to use, and has been shown to improve communication, to be a preferred modality of care, and to be especially useful when providing patient education [32]. Similar to our study, previous studies have shown high levels of patient satisfaction with video conferencing, access, time and cost savings, patient comfort and technology use [33]. Additionally, there was satisfaction in the areas of confidentiality, communication between the parties, and thoroughness of patient assessment. Despite these benefits, virtual care can have its limitations. Similar to our study, where some caregivers expressed preference for in-person assessment in order to have that personal connection, other studies have shown the absence of physical examination to be a source of dissatisfaction for some patients [33]. Another limitation is that all communities/caregivers may not have access to dependable connectivity, equipment, or knowledge to access. With these limitations in mind, it is necessary to consider patient or caregiver preferences and comfort when deciding if all or parts of the FASD evaluation process will be completed virtually. It is important that caregivers and patients are supported and well-informed regarding the virtual process from initial referral through to post-debrief steps.

A limitation specific to this proposed model was the reliance on a support worker to ensure assessments, and support was available to caregivers. This means that a community support person may need to travel to support the family/patient as a trained individual is required to be present with the patient during testing and for photographic and physical measurements. Regarding assessment, an OT assessment and a physical exam cannot be completed, while only a limited neurological examination is feasible. Not all language tests have been standardized using a virtual presentation. Other limitations include children who may be very active or are uncooperative and may not be engaged as easily. This, however, may not be different from in-person assessment.

Further research and studies are required to develop guidelines and best practices to increase success in the emerging field of using virtual models for FASD assessment. There is a need for further research to understand the full benefits of using digital or virtual models, as well as the need to understand the challenges and barriers associated with this technology.

## 5. Conclusions

Assessing and diagnosing FASD through a virtual model could be an alternative to in-person assessments when necessary. It has multiple benefits, especially for families in rural and remote settings where access to a diagnostic team might not always be convenient and feasible. The use of this virtual model was favorable to clinicians and teams with experience in FASD diagnosis and assessment as well as to caregivers of children being assessed for FASD.

## Figures and Tables

**Figure 1 children-10-00196-f001:**
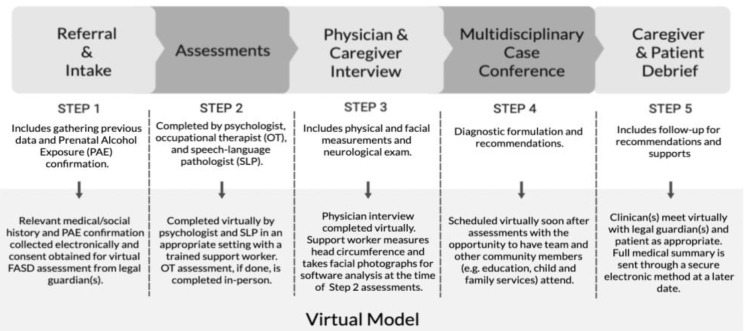
The in-person and virtual diagnosis and assessment model of Fetal Alcohol Spectrum Disorder in children.

**Figure 2 children-10-00196-f002:**
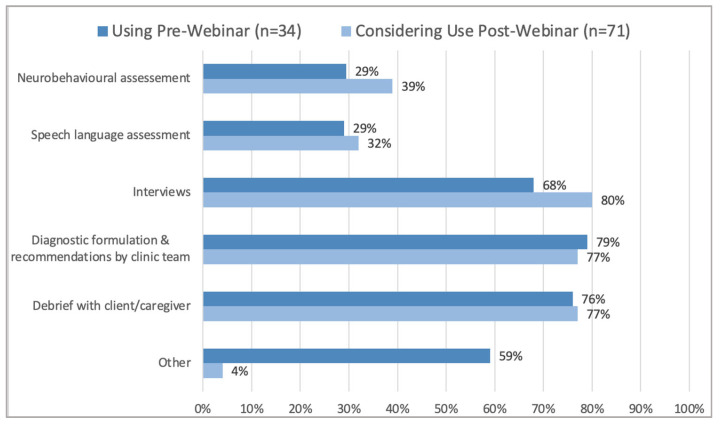
Parts of the assessment or diagnostic clinic process being used pre-webinar and considered post-webinar by FASD diagnostic teams.

**Figure 3 children-10-00196-f003:**
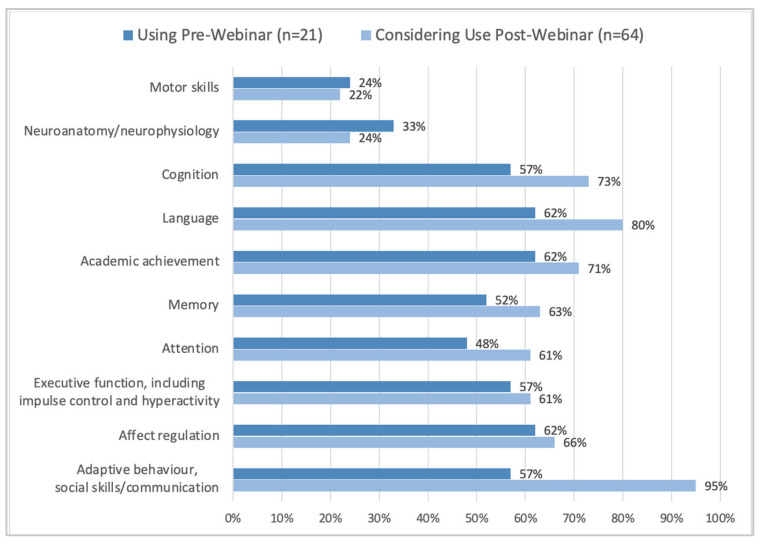
Parts of the neurodevelopmental assessments being used pre-webinar and considered post-webinar by the FASD diagnostic teams.

**Figure 4 children-10-00196-f004:**
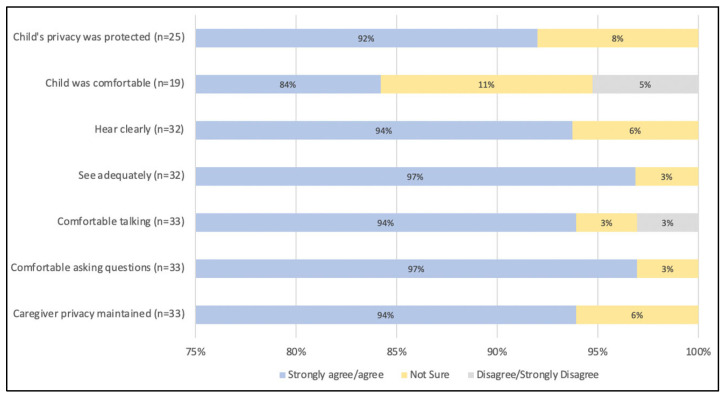
Caregiver experiences with a virtual diagnosis and assessment model of Fetal Alcohol Spectrum Disorder in children.

## Data Availability

Webinar and caregiver survey data presented in this study are available on request from the corresponding author.

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
