# Peer review of "Development and Evaluation of a Virtual Model for Fetal Alcohol Spectrum Disorder (FASD) Assessment and Diagnosis in Children: A Pilot Study"

_children, 2023, doi:10.3390/children10020196_

Round 1

Reviewer 1 Report

The manuscript describes a pilot study on Virtual Model for Fetal Alcohol Spectrum Disorder (FASD) Assessment and Diagnosis in Children. The proposed Virtual Model for diagnosing and assessing FASD is developed and validated for virtual use. The use of this model is surveyed by national and international FASD diagnostic and assessment teams and piloted in practice. This virtual model was favorable to clinicians and teams with experience in FASD diagnosis and assessment and caregivers of children being assessed for FASD. The virtual model appears helpful for some evaluations, such as cognition, language, academic achievement, memory, attention, and adaptive behaviors. In addition, it looks very promising with caregivers. Therefore, this model provides an evidence-based alternative to in-person assessments and guides pediatric FASD assessment and diagnostic teams, especially in remote and rural areas. Also, assessing and diagnosing FASD through a virtual model could be an alternative to in-person assessments when access to a diagnostic team might not always be convenient and feasible. The manuscript is well-written. Including a schematic diagram of the virtual model would be helpful. 

Author Response

Thank you for your comments and review of this paper. We agree that a schematic diagram of the virtual model would be helpful to readers and have updated our Figure 1 to show the elements of a virtual model as attached. 

Reviewer 2 Report

Overall I found this to be a good, objective study on virtual telehealth for the FASD diagnostic process. I am excited when we can present data to professionals and to families to help us understand how to serve the population of children that need FASD assessments. ‘

I found the data to be very reasonable and consistent with what we see in our practice as well. I’m very impressed that Canada has so many FASD evaluation clinics and that you have an entire team assessing kids. This is extremely rare to have a whole team together in the USA. 

In the spirit of brevity, I think this is a good paper and I agree with the methods and conclusions which are extremely consistent with what we are doing at our site. The only thing I would say or maybe add in a little stronger is the fact that subjectively (and objectively I guess) we have found that people don’t actually schedule telehealth evals for the whole FASD evaluation. I live in a very cold place and have had 95% of my patients cancel in 4 clinics this past month when we had blizzards or ice storms even though we offered to convert it to telehealth. So the people that signed up to be a part of your study might have been a biased population that was more comfortable with telehealth from the start- we have found that people will literally drive hours to see us in person.

  • I think maybe in the USA at least people are burned out on telehealth

  •  I would imagine the lengthy neuropsych part online might be difficult for them to focus’

  • Limitation that we also have encountered is that the patient still has to go “somewhere” to get the facial measurements. So they figure they might as well just come the whole way and do it in person

  • Wondering if they can reference how long each team takes with the patient. In our clinic we are an OT, psychologist and physician all working together. We have 1 hour to do our screening. Neuropsych is a different area and they are way harder to get in to so it isn’t done the same day. Feedback is written and then a feedback session is given by neuropsych (again bc they are almost always after us bc of their backlog). The reason I say that is because one of the points of articles and methods is to be able to disseminate or have other sites do what you are proposing. In canada where you have universal healthcare, I can imagine  you are able to spend as much time with the patient as you would like to- (please forgive me if I’m assuming wrong). Here in USA we literally have 3 professionals all listening, observing and giving unifying feedback all within an hour. We had to raise philanthropic funds and grant funding to stay alive to even do that and keep it moving. So feasibility in other countries might be difficult (social worker, reimbursement) 

At any rate, I agree with publication I might just acknowledge a little more some of the  possible solutions or even just acknowledgement of the difficulty of this model and maybe that patients don’t actually want telehealth as a little more emphasized

Author Response

Thank you for taking the time to review and comment on our manuscript. Please see attachment for our responses.
